# U-shaped association between serum triglyceride levels and mortality among septic patients: An analysis based on the MIMIC-IV database

**Min Xiao**[1☯], **Hongbin Deng**[2☯], **Wenjian Mao**[1], **Yang Liu**[2], **Qi Yang**[3], **Yuxiu Liu**[4]*, **Jiemei Fan**[1]*, **Weiqin Li**[1]*, **Dadong Liu**[1,5]*

1 Department of Critical Care Medicine, Jinling Hospital, Medical School of Nanjing Medical University, Nanjing, China, 2 Department of Critical Care Medicine, Jinling Hospital, Medical School of Nanjing University, Nanjing, China, 3 Department of Critical Care Medicine, Jinling Hospital, Nanjing, China, 4 Department of Biostatistics, School of Public Health, Nanjing Medical University, Nanjing, China, 5 Department of Digestive Disease Institute of Jiangsu University, Affiliated Hospital of Jiangsu University, Zhenjiang, People's Republic of China

☯ These authors contributed equally to this work.
* 583037931@qq.com (DL); liweiqindr@nju.edu.cn (WL); Liu_yuxiu@163.com (YL); aiguoshu3@163.com (JF)

**Data Availability Statement:** All relevant data are within the paper and its Supporting Information files.

## Abstract

### Background

Sepsis is characterized by upregulated lipolysis in adipose tissue and a high blood triglyceride (TG) level. It is still debated whether serum TG level is related to mortality in septic patients. The aim of this study is to investigate the association between serum TG level and mortality in septic patients admitted to the intensive care unit (ICU).

### Methods

Data from adult septic patients (≥18 years) admitted to the ICU for the first time were obtained from the Multiparameter Intelligent Monitoring in Intensive Care IV (MIMIC-IV) database. The patients' serum TG levels that were measured within the first week after ICU admission were extracted for statistical analysis. The endpoints were 28-day, ICU and in-hospital mortality.

### Results

A total of 2,782 septic patients were included. Univariate analysis indicated that the relationship between serum TG levels and the risk of mortality was significantly nonlinear. Both the Lowess smoothing technique and restricted cubic spline analyses revealed a U-shaped association between serum TG levels and mortality among septic patients. The lowest mortality rate was associated with a serum TG level of 300–500 mg/dL. Using 300 ∼ 500 mg/dL as the reference range, we found that both hypo-TG (<300 mg/dL) and hyper-TG (≥500 mg/dL) were associated with increased mortality. The result was further adjusted by Cox regression with and without the inclusion of some differential covariates.

**Funding:** This study was supported by the National Natural Science Foundation of China (No. 82202389, No. 81870441, and No. 82070669). The sponsor had no role in the design of the study, the collection and analysis of the data, or the preparation of the manuscript.

**Competing interests:** The authors have declared that no competing interests exist.

## Conclusions

There was a U-shaped association between serum TG and mortality in septic ICU patients. The optimal concentration of serum TG levels in septic ICU patients is 300–500 mg/dL.

## Introduction

Sepsis is an inflammatory disease characterized by life-threatening organ dysfunction that is caused by a dysregulated immune response to infection [1]. Despite significant progress in understanding the underlying mechanisms of the pathogenesis of sepsis and devising clinical therapeutic interventions, sepsis remains a major disease, affecting over 30 million people worldwide every year [2–4]. In the US, the estimated national weighted incidence of sepsis is 6%, and its in-hospital mortality rate is 15.6% [5]. A recent national cross-sectional survey revealed that sepsis affects one-fifth of all patients admitted to ICUs in mainland China, and the 90-day mortality rate for sepsis is 35.5% [6]. One of the reasons for the high mortality rate of sepsis is that current therapeutic approaches, which focus on anti-inflammation, hemodynamic resuscitation and organ support, disregard metabolic disorders in sepsis [7–9].

Alterations in lipid metabolism are typical metabolic disorders that occur during sepsis [10]. Sepsis upregulates lipolysis in adipose tissue [11]. As a result, septic patients have higher levels of serum triglycerides (TGs) in the blood [12]. Many studies have attempted to investigate the relationship between TG levels and mortality in patients with sepsis. Some studies have found no correlation between TG levels and mortality in sepsis patients [13–16]. Some studies have found a negative correlation [16–18], and others have found a positive one [19]. Thus, it is still unclear whether serum TG level is related to mortality in septic patients.

Here, a total of 2,782 septic patients included in an online public database called "Multiparameter Intelligent Monitoring in Intensive Care IV" (MIMIC-IV) were studied. We found a U-shaped association between serum TG level and the risk of mortality. The lowest risk of mortality was associated with a serum TG level of 300–500 mg/dL.

## Methods

### Database profile

This retrospective observational study involved septic patients included in the MIMIC IV (V.1.0) database. The MIMIC IV database was developed and is maintained by the Massachusetts Institute of Technology (MIT) Laboratory for Computational Physiology. It contains the medical data of more than 257,366 patients admitted to ICUs and emergency departments between 2008 and 2019 at the Beth Israel Deaconess Medical Center [20]. It is publicly accessible to researchers who have completed the 'protecting human subjects' training course of the National Institutes of Health (NIH).

### Ethical statement

The establishment of the MIMIC-IV database was approved by the Massachusetts Institute of Technology (Cambridge, MA) and Beth Israel Deaconess Medical Center (Boston, MA), and consent was obtained for the original data collection. The authors are accountable for all aspects of the work and ensure that questions related to the accuracy or integrity of any part of the work were appropriately investigated and resolved. The study adhered to the Declaration of Helsinki. The requirement for patient informed consent was waived due to the study design.

## Patient inclusion

Adult septic patients from the MIMIC IV (V.1.0) database were included in this study. The inclusion criteria were as follows: (1) 18 years old or older; (2) diagnosis of sepsis in line with the sepsis 3.0 criteria [21]; (3) admission to the ICU for the first time; and (4) TG level testing within the first week after ICU admission. According to the sepsis 3.0 criteria, the diagnosis of sepsis requires the presence of infection (suspected infection) and organ dysfunction. Infection means that the patient has clear etiological evidence, and suspected infection is defined as undergoing any culture and at least two doses of antimicrobials. Organ dysfunction is defined as an acute change in total Sequential Organ Failure Assessment (SOFA) score of 2 points or higher consequent to the infection. The TG levels were recorded every day if the patients underwent TG testing within the first week after ICU admission. The exclusion criteria were as follows: (1) less than 18 years old; (2) pregnant or lactating or in a permanent vegetative state; or (3) presence of a communicable disease, such as AIDS.

## Data extraction

Data were extracted by Deng, who completed the online training course of the NIH (certification number: 39921130). The extracted parameters included age, sex, weight, height, SOFA score, comorbidities, heart rate, mean arterial pressure (MAP), blood glucose level, lactate level, white blood cell (WBC), neutrophil, and platelet counts, hematocrit, hemoglobin, blood urea nitrogen (BUN), creatinine, alanine transaminase (ALT), aspartate aminotransferase (AST), albumin, total bilirubin levels, and renal replacement therapy (RRT) within 7 days. Comorbidities included hypertension, diabetes, hyperlipidemia, chronic pulmonary disease (CPD), myocardial infarct (MI), congestive heart failure (CHF), atherosclerosis, vascular disease, liver disease, renal disease, hypothyroidism, pancreatitis, and tumor. We also collected the patients' serum TG levels measured within the first week after ICU admission. Then, both the maximum value of TG ($TG_{max}$) and minimum value of TG ($TG_{min}$) were selected for statistical analysis. The endpoints were 28-day, ICU and in-hospital mortality. Data extraction was performed using PostgreSQL tools V.1.12.3.

## Statistical analysis

All data were statistically analyzed using SPSS 20.0 and R 4.1.2 software. Continuous variables are presented as medians (interquartile ranges, IQRs). Differences were analyzed by the Mann-Whitney U test (for 2 groups) or Kruskal–Wallis H test (for 3 groups). Categorical variables are presented as frequencies (percentages), and their differences were analyzed by chi-square tests. In total, less than 10% of the data were missing for the continuous variables of height and weight. The missing data were imputed using multiple imputation with chain equations. Then, the body mass index (BMI) was calculated by the formula BMI = weight/(height * height). The number of participants with missing data for each variable of interest is shown in **S1 Fig**.

The relationship among the septic patients was further adjusted by restricted cubic splines with or without some differential covariates. The reference was set at 150 mg/dL (the upper limit of the reference value of serum TGs). Cox regression was used to explore the relationship between TG level and mortality in septic patients. Variables were selected for inclusion in the models based on statistical significance in the univariate analyses, biological plausibility and known associations with in-hospital mortality. Kaplan–Meier (K-M) survival curves were constructed. $P < 0.05$ was considered statistically significant.

## Results

### Low serum TG values are associated with higher mortality

In total, 11,263 adult septic patients were included in the MIMIC-IV database. After a rigorous screening process, 2,782 septic patients were included in this study (**Fig 1**). Their median age (IQR) was 63.9 (51.8, 74.5) years, and female patients accounted for 40.2% of patients. Among the included patients, 605 (21.7%) died within 28 days, 487 (17.5%) died in the ICU, and 651 (23.4%) died in the hospital. We found that the serum TG levels of the patients who died within 28 days were lower than those of the survivors (**Table 1**). Similar results were found in the patients who died in the ICU (**S1 Table**) and in the hospital (**S2 Table**). These data indicated that a low serum TG value was associated with higher mortality.

In total, 1,444 (51.9%) patients had serum $TG_{max}$ levels of less than 150 mg/dL (clinically normal TG level), 1,118 (40.2%) patients had serum $TG_{max}$ levels between 150 mg/dL and 500 mg/dL, and the remaining 220 (7.9%) patients had serum $TG_{max}$ levels above 500 mg/dL. We calculated mortality in the septic patients in the 3 groups defined by the different $TG_{max}$ levels. The results showed that patients with serum $TG_{max}$ levels of $150 \sim 500$ mg/dL had the lowest mortality (**Table 2**). Similar results were found in serum $TG_{min}$ levels (**Table 2**).

All of these results indicated a nonlinear association between serum TG levels and mortality among septic patients.

### U-shaped association between serum TG and mortality

By using the Lowess smoothing technique, we preliminarily explored the relationship between serum TG levels and mortality in patients with sepsis. Three models yielded nonlinear relationships, with the lowest mortality rate at a serum TG level of $300 \sim 500$ mmol/L (**Figs 2A–2C** and **S2A-S2C**).

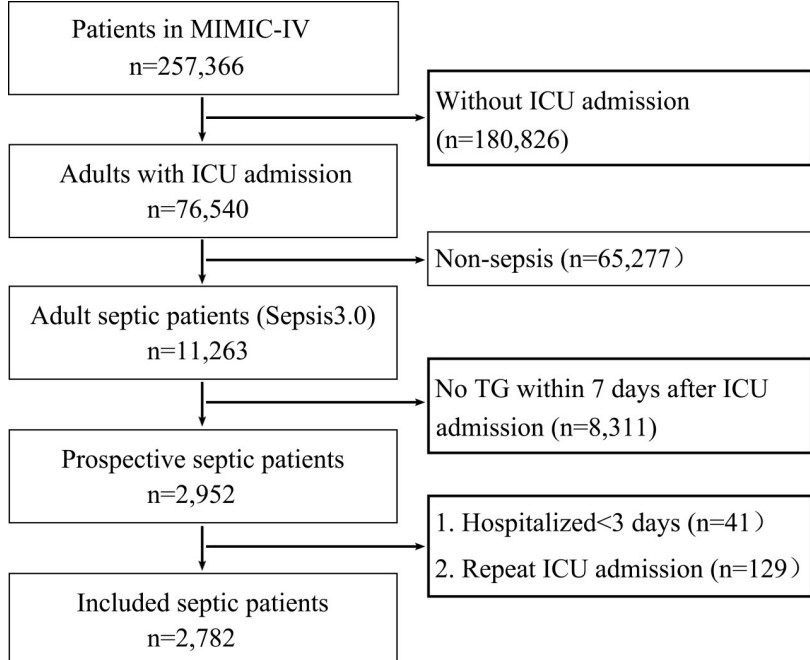

**Fig 1. Flowchart of septic patient inclusion.**

**Table 1. Characteristics of survivors and non-survivors within 28 days.**

| Variables | Total (n = 2782) | Survivors (n = 2177) | Non-survivors (n = 605) | *p* value |
|---|---|---|---|---|
| Age (years) | 63.9 (51.8, 74.5) | 63.2 (50.5, 73.8) | 61.2 (55.9, 77.3) | <**0.001** |
| Female (n (%)) | 1119 (40.2) | 877 (40.3) | 232 (38.4) | 0.389 |
| BMI (kg/m$^2$) | 28.4 (24.4, 33.8) | 28.4 (24.5, 33.7) | 28.2 (24.1, 33.9) | <**0.001** |
| SOFA score | 8.0 (5.0, 12.0) | 7.0 (5.0, 11.0) | 11.0 (7.0, 14.0) | <**0.001** |
| **Comorbidities** | | | | |
| Hypertension (n (%)) | 1075 (38.6) | 851 (39.1) | 224 (37.0) | 0.356 |
| Diabetes (n (%)) | 834 (30.0) | 644 (29.6) | 190 (31.4) | 0.387 |
| Hyperlipidemia (n (%)) | 890 (32.0) | 689 (31.7) | 201 (33.2) | 0.463 |
| CPD (n (%)) | 754 (27.1) | 574 (26.4) | 180 (29.8) | 0.097 |
| MI (n (%)) | 516 (18.6) | 393 (18.1) | 123 (20.3) | 0.202 |
| CHF (n (%)) | 902 (32.4) | 670 (30.8) | 232 (38.4) | <**0.001** |
| Atherosclerosis (n (%)) | 336 (32.4) | 247 (11.4) | 89 (14.7) | **0.025** |
| Vascular disease (n (%)) | 978 (35.2) | 751 (34.5) | 227 (37.5) | 0.168 |
| Liver disease (n (%)) | 600 (21.6) | 420 (19.3) | 180 (29.8) | <**0.001** |
| Renal disease (n (%)) | 593 (21.3) | 434 (19.9) | 159 (26.3) | **0.001** |
| Hypothyroidism (n (%)) | 348(12.5) | 268(12.3) | 80(13.2) | 0.548 |
| Pancreatitis (n (%)) | 202(7.3) | 170(7.8) | 32(5.3) | **0.035** |
| Tumor (n (%)) | 332 (11.9) | 228 (10.47) | 104 (17.2) | <**0.001** |
| **During the first 24 hours after ICU admission** | | | | |
| Heart rate (beat/min) | 109.0 (95.0, 124.0) | 108.0 (95.0, 124.0) | 111.0 (96.0, 126.0) | 0.480 |
| MAP (mmHg) | 59.0 (53.00, 66.0) | 59.0 (53.0, 66.0) | 58.0 (52.0, 63.0) | <**0.001** |
| Blood glucose (mg/dL) | 167.0 (131.0, 225.0) | 165.0 (130.0, 222.0) | 177.0 (137.00, 241.00) | **0.003** |
| Lactate (mmol/L) | 2.2 (1.6, 3.1) | 2.2 (1.5, 2.9) | 2.2 (1.9, 4.5) | <**0.001** |
| WBC (×10$^9$/L) | 13.9 (10.1, 19.0) | 13.7 (9.9, 18.6) | 15.5 (11.2, 21.4) | <**0.001** |
| Platelet (×10$^9$/L) | 209.0 (148.0, 281.0) | 212.0 (151.0, 285.0) | 197.0 (132.0, 267.0) | **0.001** |
| Hematocrit (%) | 35.5 (30.5, 40.9) | 35.6 (30.7, 41.0) | 34.5 (29.3, 40.5) | <**0.001** |
| Hemoglobin (g/L) | 11.60 (9.90, 13.5) | 11.7 (10.0, 13.5) | 11.1 (9.5, 13.2) | <**0.001** |
| BUN (mg/dL) | 24.0 (16.0, 39.0) | 22.0 (15.0, 36.0) | 30.0 (20.0, 51.0) | <**0.001** |
| Creatinine (mg/dL) | 1.2 (0.9, 2.0) | 1.2 (0.8, 1.9) | 1.5 (1.0, 2.4) | <**0.001** |
| ALT (IU/L) | 31.0 (21.0, 50.0) | 31.0 (21.0, 49.0) | 31.0 (21.0, 59.0) | 0.324 |
| AST (IU/L) | 47.0 (32.0, 79.0) | 47.0 (31.0, 73.0) | 47.0 (37.0, 104.0) | <**0.001** |
| Albumin (g/dL) | 3.3 (3.0, 3.5) | 3.3 (3.1, 3.5) | 3.3 (2.8, 3.4) | **0.001** |
| Total bilirubin (mg/dL) | 0.8 (0.5, 1.4) | 0.8 (0.5, 1.2) | 0.8 (0.6, 2.2) | <**0.001** |
| RRT within 7 days (n (%)) | 443 (15.9) | 288 (13.2) | 155 (25.6) | <**0.001** |
| TG$_{max}$ level (mg/dL) | 145.0 (93.0, 249.0) | 148.0 (95.0, 254.0) | 136.0 (88.0, 227.0) | **0.013** |
| TG$_{min}$ level (mg/dL) | 132.5 (88.0, 207.0) | 135.0 (90.0, 211.0) | 123.0 (84.0, 190.0) | **0.001** |

**Note:** Continuous variables (age, BMI, MAP, *et al.*) were presented as median (IQR). Categorical variables (hypertension, CPD, MI, *et al.*) were presented as frequencies (percentages). The differences between survivors and non-survivors were analyzed by Mann-Whitney U test, and Chi-square. BMI: body mass index; SOFA: sequential organ failure assessment; CPD: chronic pulmonary disease; MI: myocardial infarct; CHF: congestive heart failure; MAP: mean arterial pressure; WBC: white blood cell; BUN: blood urea nitrogen; ALT: alanine transaminase; AST: aspartate aminotransferase; RRT: renal replacement therapy; TG$_{max}$: maximum value of triglycerides; TG$_{min}$: minimum value of triglycerides.

The relationship between serum TG levels and risk of mortality was further adjusted by restricted cubic splines with or without some differential covariates. When the serum TG$_{max}$ level was examined as a continuous variable in relation to mortality, a significant nonlinear pattern ($p < 0.05$) was demonstrated, which resulted in a U-shaped curve (**Fig 2D–2F**). The

**Table 2. Mortality of different serum TG groups (clinical boundary values).**

| Variables | Total | < 150 mg/dL | 150–500 mg/dL | > 500 mg/dL | *p* value |
|---|---|---|---|---|---|
| **Serum TG$_{max}$ levels** | 2782 | 1444 (51.9) | 1118 (40.2) | 220 (7.9) | |
| 28-day mortality (n(%)) | 605 (21.7) | 341 (23.61) | 211 (18.87) [*] | 53 (24.09) [#] | **0.011** |
| ICU mortality (n(%)) | 487 (17.51) | 270 (18.70) | 169 (15.12) [*] | 48 (21.82) [#] | **0.013** |
| In-hospital mortality (n(%)) | 651 (23.40) | 363 (25.14) | 229 (20.48) [*] | 59 (26.82) [#] | **0.010** |
| **Serum TG$_{min}$ levels** | 2782 | 1627 (58.5) | 1047 (37.6) | 108 (3.9) | |
| 28-day mortality (n(%)) | 605 (21.7) | 386 (23.7) | 198 (18.9) [*] | 21 (19.4) | **0.011** |
| ICU mortality (n(%)) | 487 (17.51) | 313 (19.2) | 153 (14.6) [*] | 21 (19.4) | **0.008** |
| In-hospital mortality (n(%)) | 651 (23.40) | 415 (25.5) | 211 (20.2) [*] | 25 (23.1) | **0.006** |

**Note:** Data were presented as frequencies (percentages). The differences between survivors and non-survivors were analyzed by Chi-square.

[*]Compared with < 150 mg/dL, $P < 0.05$

[#]Compared with 150–500 mg/dL, $P < 0.05$. TG$_{max}$: maximum value of triglycerides; TG$_{min}$: minimum value of triglycerides; ICU: Intensive Care Unit.

lowest 28-day mortality rate was observed with a TG$_{max}$ level of 396.5 mg/dL. The lowest ICU mortality rate was observed with a TG$_{max}$ level of 459.6 mg/dL, and the lowest in-hospital mortality rate was observed with a TG$_{max}$ level of 410.5 mg/dL. Similar results were found in serum TG$_{min}$ levels (**S2D-S2F Fig**). All of these results indicate a U-shaped association between serum TG levels and mortality among septic patients.

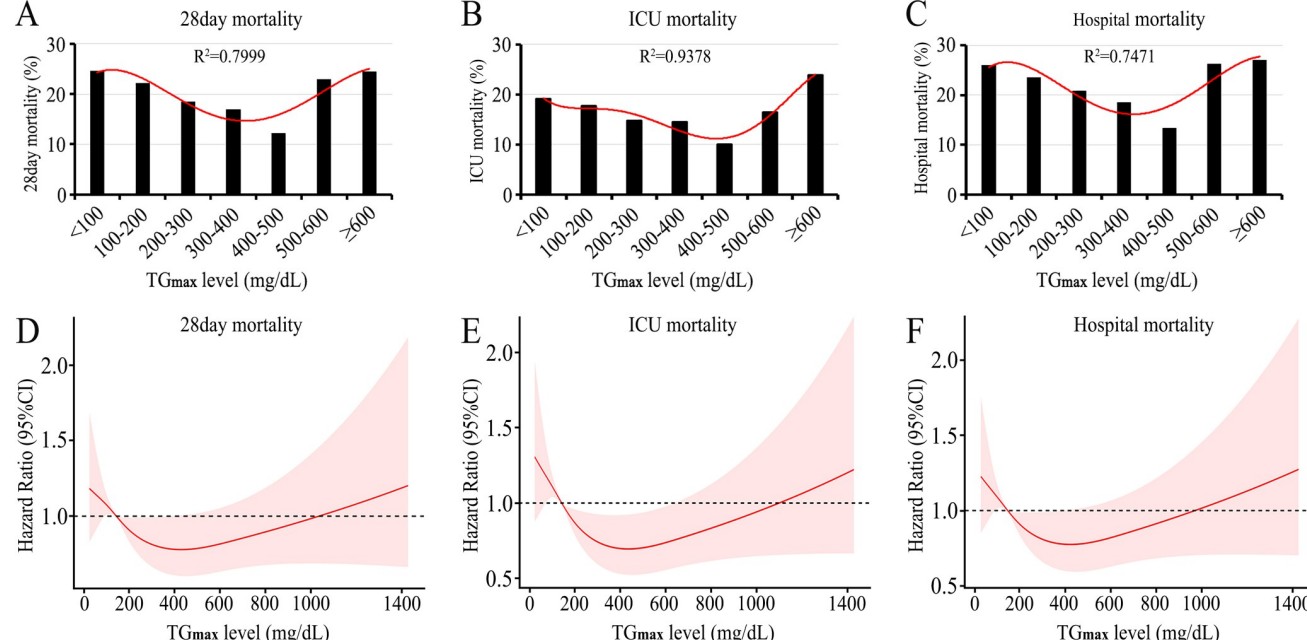

**Fig 2. U-shaped association between serum TG$_{max}$ levels and mortality among septic patients.** A. Lowess smoothing technique revealed that the lowest 28-day mortality (12.22%) was associated with TG$_{max}$ levels of 400–500 mg/dL. B. Lowess smoothing technique revealed that the lowest ICU mortality (10.00%) was associated with TG$_{max}$ levels of 400–500 mg/dL. C. Lowess smoothing technique revealed that the lowest hospital mortality (13.33%) was associated with a TG$_{max}$ level of 400–500 mg/dL. D. Restricted cubic splines revealed that the relationship between serum TG$_{max}$ level and risk of 28-day mortality was significantly nonlinear ($p = 0.039$), resulting in a U-shaped association. The lowest hospital mortality (HR: 0.78, 95% CI: 0.61, 0.99; $p <0.05$) was associated with a TG$_{max}$ level of 396.5 mg/dL. E. Restricted cubic splines revealed that the relationship between serum TG$_{max}$ level and risk of ICU mortality was significantly nonlinear ($p = 0.002$), resulting in a U-shaped association. The lowest ICU mortality (HR: 0.70, 95% CI: 0.53,0.93; $p <0.05$) was associated with a TG$_{max}$ level of 459.6 mg/dL. F. Restricted cubic splines revealed that the relationship between serum TG$_{max}$ levels and the risk of hospital mortality was significantly nonlinear ($p = 0.013$), resulting in a U-shaped association. The lowest hospital mortality (HR: 0.78, 95% CI: 0.66, 0.99; $p <0.05$) was associated with a TG$_{max}$ level of 410.5 mg/dL. TG$_{max}$: the maximum value of serum triglyceride; ICU: intensive care unit.

In this study, both Lowess smoothing technique and restricted cubic splines indicated that the boundaries of TG levels were less than 300 mg/dL, 300–500 mg/dL and greater than 500 mg/dL. However, clinically normal TG levels are less than 150 mg/dL. To investigate the effect of mildly elevated TG levels on the mortality of septic patients, a total of 2,272 septic patients who had TG levels of less than 300 mg/dL were divided into a normal group ($TG_{max}$ less than 300 mg/dL) and an elevated group ($TG_{max}$: 300–500 mg/dL). We found that there was no significant difference in mortality (including 28-day mortality, ICU mortality and in-hospital mortality) between the 2 groups (**S3 Table**). Therefore, we artificially divided TG values into 3 boundaries: less than 300 mg/dL, 300 to 500 mg/dL, and greater than 500 mg/dL.

## Both hypo-TG and hyper-TG were associated with increased mortality

According to the U-shaped association, $TG_{max}$ values were selected and categorized into 3 groups: hypo-$TG_{max}$ ($< 300$ mg/dL), moderate-$TG_{max}$ ($300 \sim 500$ mg/dL), and hyper-$TG_{max}$ ($\geq 500$ mg/dL). The primary clinical outcomes unadjusted for covariates for the 3 $TG_{max}$ groups are presented in **Table 3**. The results showed that hypo-$TG_{max}$ or hyper-$TG_{max}$ was associated with increased mortality. To explore the effect of TG levels on mortality, moderate-$TG_{max}$ was selected as the reference group for the Cox regression models. The results showed that both hypo-$TG_{max}$ and hyper-$TG_{max}$ were associated with increased mortality among septic patients (**Table 4**). Survival analysis (Kaplan–Meier) further indicated that patients with hypo-$TG_{max}$ and hyper-$TG_{max}$ had higher mortality rates (**Fig 3**).

## Discussion

The amount of TGs is directly proportional to the severity of sepsis [22], but evidence of how TG levels affect the prognosis of sepsis is limited. In this study, 2,782 septic patients with serum TG levels measured within the first week after ICU admission were selected from the MIMIC-IV database. Our results first revealed a U-shaped association between serum TG levels and mortality among ICU-admitted septic patients. The lowest mortality rate was associated with a serum TG level of 300–500 mg/dl. Both hypo-TG ($<$300 mg/dL) and hyper-TG ($\geq$500 mg/dL) were associated with increased mortality among septic patients. To the best of our knowledge, this study is the first retrospective observational study to suggest a link between serum TG (within the first week after ICU admission) imbalance and mortality among septic patients.

TGs are biomolecules that were discovered many years ago. They are derived from glycerol and three fatty acids and have key roles in the storage and transportation of fatty acids in cells and plasma [23–25]. Under normal physiological conditions, a small amount of TGs ($<$150 mg/dL) can be detected in peripheral blood. Sepsis is accompanied by severe metabolic alterations. In 1992, Kenneth et al. demonstrated that lipopolysaccharide (LPS) can rapidly induce hypertriglyceridemia [26]. They found that low doses of LPS can stimulate hepatic TG production, while high doses of LPS may reduce TG hydrolysis in cells and plasma. Further study demonstrated a shift from glucose metabolism to lipid metabolism in LPS-challenged mice [27]. They found that in sepsis, lipids are transported to the liver and serve as the dominant energy source for septic mice. In addition, a marked elevation in serum TG levels has been observed in septic models and patients [11, 17, 18, 28]. Furthermore, a recent published study reported that baseline TG levels were significantly higher in patients with septic shock than in patients with sepsis [29]. All of these results indicate that sepsis is usually accompanied by elevated serum TG levels, and TGs are the main energy supply substrate for sepsis patients. In this study, we found that nearly half of the septic patients had serum TG levels at the limit of the reference value (150 mg/dL), and these patients had higher mortality rates. Moreover,

**Table 3. Both hypo-TG$_{max}$ and hyper-TG$_{max}$ are associated with higher mortality.**

| Variables | Hypo-TG$_{max}$ (n = 1,444) | Moderate-TG$_{max}$ (n = 1,118) | Hyper-TG$_{max}$ (n = 220) | p value |
|---|---|---|---|---|
| Age (years) | 65.4 (53.9, 76.3) | 56.5 (42.7 66.5) | 54.75 (42.5, 66.2) | < **0.001** |
| Female (n (%)) | 935 (41.2) | 99 (34.1) | 85 (38.6) | 0.063 |
| BMI (kg/m$^2$) | 27.8 (23.9, 32.8) | 30.9 (26.5, 37.1) | 32.4 (27.7, 37.1) | < **0.001** |
| SOFA score | 8.0 (5.0, 11.0) | 9.0 (6.0, 13.0) | 11.0 (7.0, 13.0) | < **0.001** |
| **Comorbidities** | | | | |
| Hypertension (n (%)) | 878 (38.6) | 121 (41.7) | 76 (34.6) | 0.257 |
| Diabetes (n (%)) | 660 (29.15) | 92 (31.7) | 82 (37.3) | **0.031** |
| Hyperlipidemia (n (%)) | 743 (32.7) | 80 (27.6) | 67 (30.5) | 0.187 |
| CPD (n (%)) | 609 (26.8) | 90 (31.0) | 55 (25.0) | 0.239 |
| MI (n (%)) | 455 (20.0) | 34 (11.7) | 27 (12.3) | < **0.001** |
| CHF (n (%)) | 790 (34.8) | 72 (24.8) | 40 (18.2) | < **0.001** |
| Atherosclerosis (n (%)) | 266 (11.7) | 38 (13.1) | 32 (14.6) | 0.398 |
| Vascular disease (n (%)) | 852 (37.5) | 65 (22.4) | 61 (27.7) | < **0.001** |
| Liver disease (n (%)) | 493 (21.7) | 55 (19.0) | 52 (23.6) | 0.419 |
| Renal disease (n (%)) | 491 (21.6) | 53 (18.3) | 49 (22.3) | 0.399 |
| Tumor (n (%)) | 266 (11.7) | 39 (13.5) | 27 (12.3) | 0.681 |
| **During the first 24 hours after ICU admission** | | | | |
| Heart rate (beat/min) | 108.0 (94.0, 123.0) | 114.5 (100.0, 130.3) | 117.0 (101.0, 128.5) | < **0.001** |
| MAP (mmHg) | 59.0 (53.0, 66.0) | 59.0 (54.0, 65.0) | 58.0 (52.0, 65.0) | 0.501 |
| Blood glucose (mg/dL) | 164.0 (130.0, 219.0) | 175.0 (133.8, 257.3) | 182.5 (141.3, 266.5) | < **0.001** |
| Lactate (mmol/L) | 2.2 (1.6, 3.0) | 2.2 (1.5, 3.8) | 2.2 (1.4, 4.4) | 0.567 |
| WBC (×10$^9$/L) | 13.9 (9.9, 18.7) | 14.8 (10.6, 20.1) | 13.9 (10.7, 20.9) | **0.026** |
| Platelet (×10$^9$/L) | 210.0 (149.3, 280.0) | 216.0 (151.8, 296.0) | 191.5 (130.0, 265.8) | **0.027** |
| Hematocrit (%) | 35.5 (30.6, 41.0) | 35.5 (29.6, 41.2) | 34.7 (29.8, 40.1) | 0.223 |
| Hemoglobin (g/L) | 11.6 (9.9, 13.4) | 11.6 (9.7, 13.7) | 11.4 (9.7, 13.2) | 0.347 |
| BUN (mg/dL) | 23.0 (15.0, 38.0) | 24.0 (17.0, 46.3) | 26.0 (17.3, 51.8) | **0.004** |
| Creatinine (mg/dL) | 1.2 (0.8, 1.8) | 1.3 (0.9, 2.4) | 1.6 (1.0, 2.6) | < **0.001** |
| ALT (IU/L) | 31.0 (20.0, 47.0) | 31.0 (24.0, 65.0) | 36.5 (26.0, 98.0) | < **0.001** |
| AST (IU/L) | 47.0 (30.0, 73.0) | 47.0 (38.8, 88.0) | 47.0 (42.3, 1.5) | < **0.001** |
| Albumin (g/dL) | 3.3 (3.1, 3.6) | 3.3 (2.9, 3.5) | 3.3 (2.7, 3.3) | < **0.001** |
| Total bilirubin (mg/dL) | 0.8 (0.5, 1.3) | 0.8 (0.5, 1.5) | 0.8 (0.5, 2.0) | 0.284 |
| RRT within 7days (n (%)) | 306 (13.5) | 58 (20.0) [*] | 79 (35.9) [*#] | < **0.001** |
| 28-day mortality (n (%)) | 507(22.32) | 45(15.52) [*] | 53(24.09) [#] | **0.021** |
| ICU mortality (n (%)) | 401 (17.7) | 38 (13.1) | 48 (21.8) [#] | < **0.001** |
| Hospital mortality (n (%)) | 543 (23.9) | 49 (16.9) [*] | 59 (26.8) [#] | **0.014** |

**Note:** Hypo-TG$_{max}$: <300 mg/dL, Moderate-TG$_{max}$: 300~500 mg/dL, Hyper-TG$_{max}$ ≥500 mg/dL. Continuous variables (age, BMI, MAP, *et al.*) were presented as median (IQR). Categorical variables (hypertension, CPD, MI, *et al.*) were presented as frequencies (percentages). The difference among the 3 groups was analyzed by Kruskal-Wallis H test.

[*]Compared with Hypo-TG$_{max}$, $P < 0.05$

[#]Compared with Moderate-TG$_{max}$, $P < 0.05$. BMI: body mass index; SOFA: sequential organ failure assessment; CPD: chronic pulmonary disease; MI: myocardial infarct; CHF: congestive heart failure; MAP: mean arterial pressure; WBC: white blood cell; BUN: blood urea nitrogen; ALT: alanine transaminase; AST: aspartate aminotransferase; RRT: renal replacement therapy; ICU: Intensive Care Unit; TG: Triglycerides.

similar results were observed in patients with excessively high serum TG levels (>500 mg/dL). Furthermore, we revealed that the nonsurvivors had lower serum TG values than the survivors. These findings indicate that the serum TG level is directly proportional to the severity of sepsis but inversely proportional to the mortality rate in septic patients.

**Table 4. Cox regression based on serum TG$_{max}$ groups.**

| Models | 28-day mortality | | | ICU mortality | | | In-hospital mortality | | |
|---|---|---|---|---|---|---|---|---|---|
| | HR | 95% CI | *p* value | HR | 95% CI | *p* value | HR | 95% CI | *p* value |
| **Model 1** | | | | | | | | | |
| Hypo-TG | 1.51 | 1.11, 2.04 | 0.008 | 1.96 | 1.40, 2.74 | < 0.001 | 1.78 | 1.33, 2.39 | **< 0.001** |
| Hyper-TG | 1.63 | 1.09, 2.42 | 0.016 | 1.81 | 1.18, 2.77 | 0.007 | 1.71 | 1.17, 2.50 | **0.005** |
| **Model 2** | | | | | | | | | |
| Hypo-TG | 1.52 | 1.11, 2.07 | 0.009 | 1.90 | 1.35, 2.68 | < 0.001 | 1.71 | 1.27, 2.30 | **< 0.001** |
| Hyper-TG | 1.52 | 1.02, 2.26 | 0.039 | 1.77 | 1.16, 2.71 | 0.009 | 1.68 | 1.15, 2.46 | **0.007** |
| **Model 3** | | | | | | | | | |
| Hypo-TG | 1.43 | 1.05, 1.96 | 0.025 | 1.81 | 1.29, 2.56 | 0.001 | 1.59 | 1.18, 2.15 | **0.002** |
| Hyper-TG | 1.52 | 1.02, 2.27 | 0.040 | 1.84 | 1.20, 2.82 | 0.006 | 1.69 | 1.16, 2.48 | **0.007** |
| **Model 4** | | | | | | | | | |
| Hypo-TG | 1.46 | 1.06, 2.00 | 0.019 | 1.86 | 1.31, 2.63 | < 0.001 | 1.60 | 1.18, 2.17 | **0.002** |
| Hyper-TG | 1.57 | 1.05, 2.35 | 0.029 | 1.84 | 1.19, 2.84 | 0.006 | 1.71 | 1.16, 2.52 | **0.006** |
| **Model 5** | | | | | | | | | |
| Hypo-TG | 1.48 | 1.08, 2.04 | 0.014 | 1.90 | 1.34, 2.69 | < 0.001 | 1.62 | 1.20, 2.20 | **0.002** |
| Hyper-TG | 1.50 | 1.00, 2.25 | 0.049 | 1.79 | 1.16, 2.78 | 0.009 | 1.67 | 1.13, 2.46 | **0.009** |

**Note:** Among the 3 TG groups, moderate-TG (300 ∼ 500 mg/dL) was adopted as the reference group. Hypo-TG: <300 mg/dL, Hyper-TG ≥500 mg/dL.

**Model 1:** unadjusted model.

**Model 2:** Adjusted for age, gender, BMI and SOFA score.

**Model 3:** Adjusted for variables included in model 2 + hypertension, diabetes, hyperlipidemia, CPD, MI, CHF, atherosclerosis, vascular disease, liver disease, renal disease, and tumor.

**Model 4:** Adjusted for variables included in model 3 + heart rate, MAP, blood glucose, lactate, WBC, platelet, hematocrit, hemoglobin. BUN, creatinine, ALT, AST, albumin, and total bilirubin.

**Model 5:** Adjusted for variables included in model 4 + RRT within 7days.

The main function of TG is to store unused calories and provide the body with energy. Previous studies showed that lower serum TG levels were significantly associated with higher hospital mortality rates among septic patients [18, 24, 30–32]. Furthermore, Maile et al. reported that the pre-sepsis TG level was associated with the severity of illness, and high TG values led to higher in-hospital mortality among septic patients [19]. A recent quantitative proteomic analysis

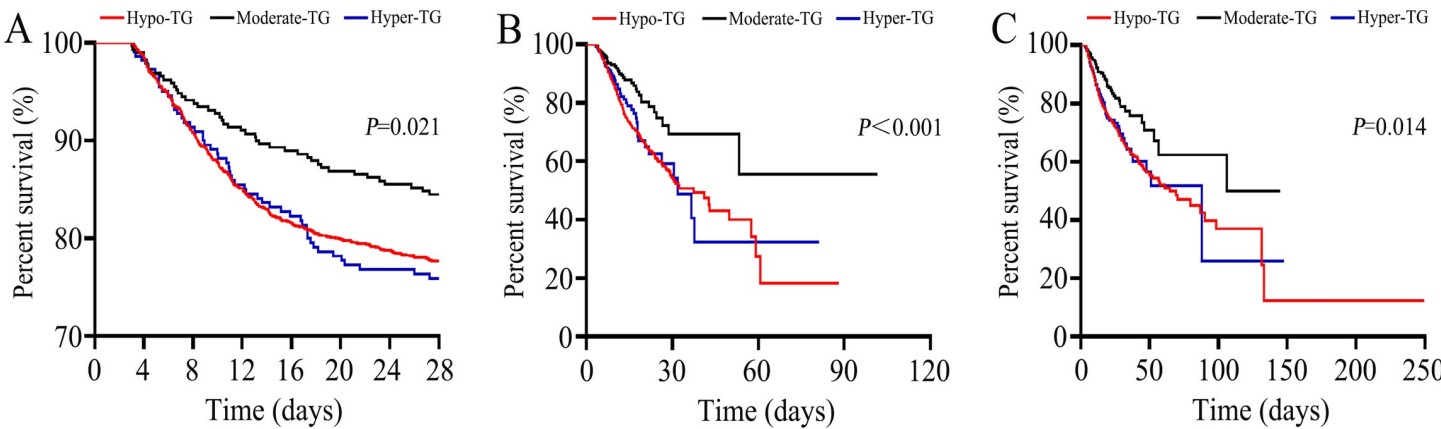

**Fig 3. Survival curve based on serum TG groups.** Both hypo-TG and hyper-TG were associated with higher 28-day mortality (A), ICU mortality (B) and hospital mortality (C) among septic patients. TG$_{max}$: the maximum value of serum triglyceride; ICU: intensive care unit.

showed that TG/cholesterol homeostasis was associated with sepsis-related acute kidney injury [33]. However, many other studies, which involved simple comparisons between survivors and nonsurvivors, reported similar TG levels in deceased patients and survivors [13–16].

Unlike previous studies, our study revealed a 'U'-shaped relationship between changes in serum TG levels and mortality among septic patients. Further analysis revealed that both hypo-TG ($<$300 mg/dL) and hyper-TG ($\geq$500 mg/dL) were associated with increased mortality. These results were similar to those of Choi et al., who reported that both hypertriglyceridemia and hypotriglyceridemia may be associated with poor early outcomes of acute ischemic stroke [34]. While the specific mechanism remains unclear, we speculate that it may be as follows. First, the body of a septic patient enters a state of stress, resulting in hypermetabolism and higher energy consumption. Therefore, an extremely low level of TGs, which serve to store energy, may lead to poor prognoses [35–37]. Second, lipid metabolism disorders may seriously damage immune function, leading to uncontrolled infection and inflammatory damage to tissues and organs [38–40]. Recent studies have pointed out that TG-based markers are associated with many inflammatory conditions including type 2 diabetes mellitus, coronary heart disease and liver steatosis [41–43]. The initial phase of sepsis is characterized by overwhelming inflammation [44]. Interestingly, a recent study revealed that a combination of bis ($\alpha$-furancarboxylato) oxovanadium and metformin ameliorates hepatic steatosis in high-fat diet-induced obese mice by alleviating hepatic inflammation and enhancing the insulin signaling pathway [45]. Thus, we speculate that TG may be associated with sepsis mortality via the inflammation pathway. This should be further investigated soon. In addition, hyper-TG is associated with elevated blood viscosity and reduced blood flow velocity [46]. As a result, hypertriglyceridemia restricts the blood supply to tissue and organs. All of these factors result in a poor prognosis in septic patients.

There are some limitations to this study. First, this is a retrospective trial, and the data in the MIMIC-IV database were not collected for this purpose. Therefore, our findings should be considered purely hypothesis-generating rather than confirmatory and need to be further validated in clinical practice. Second, the risk of sampling bias should be considered. In this study, our data were extracted from the MIMIC-IV database. The database is not a routine laboratory for sepsis management. Therefore, the act of simply having one drawn is likely to add bias to the sample. Third, the nutritional statuses of the patients before ICU admission were not clear. We extracted only height and weight parameters and calculated BMI values. No significant difference was found in BMI between the survivors and nonsurvivors. Fourth, we did not obtain data on inflammatory cytokines (including interleukin-1, interleukin-6 and tumor necrosis factor $\alpha$) that may have contributed to lipid metabolism [47]. Inflammatory cytokines were not available in the MIMIC-IV (1.0) database. Fifth, our study is a single-center study that may have potential bias. Therefore, a better-designed prospective study involving different ethnic groups is required to reconfirm the role of the TG level in the prognosis of septic patients.

## Conclusions

In conclusion, this study is the first retrospective observational study to suggest a U-shaped association between serum TG (within the first week after ICU admission) imbalance and mortality among septic patients. According to the link between serum TG (within the first week after ICU admission) imbalance and mortality among septic patients, we should maintain serum TG levels in septic patients at 300–500 mg/dL.

## Supporting information

**S1 Table. Characteristics of survivors and non-survivors within ICU.**
(DOCX)

**S2 Table. Characteristics of survivors and non-survivors in hospital.**
(DOCX)

**S3 Table. Mortality of different serum $TG_{max}$ groups (clinical boundary values).**
(DOCX)

**S1 Fig. Number of participants with missing data for each variable of interest.**
(DOCX)

**S2 Fig. U-shaped association between serum $TG_{min}$ levels and mortality among septic patients.**
(DOCX)

## Acknowledgments

We thank all the staff of the MIT Laboratory for Computational Physiology and Beth Israel Deaconess Medical Center for their work on the MIMIC IV database. We thank all the patients in the MIMIC IV database for their selfless contribution of their medical data.

## Author Contributions

**Conceptualization:** Yang Liu.

**Data curation:** Hongbin Deng.

**Formal analysis:** Dadong Liu.

**Investigation:** Qi Yang.

**Methodology:** Min Xiao.

**Project administration:** Weiqin Li.

**Resources:** Wenjian Mao.

**Supervision:** Yuxiu Liu, Jiemei Fan.

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
