## [Decision Letter · Decision Letter 0]

27 Jul 2023

PONE-D-23-19155U-Shaped association between serum triglyceride levels and mortality among septic patients: an analysis based on the MINIC-IV databasePLOS ONE

Dear Dr. Li,

Thank you for submitting your manuscript to PLOS ONE. After careful consideration, we feel that it has merit but does not fully meet PLOS ONE’s publication criteria as it currently stands. Therefore, we invite you to submit a revised version of the manuscript that addresses the points raised during the review process.

We look forward to receiving your revised manuscript.

Kind regards,

Gulali Aktas

Academic Editor

PLOS ONE

Journal Requirements:

Additional Editor Comments:

Authors are invited to revise their paper according to the suggestions of the reviewers.

Reviewers' comments:

Reviewer's Responses to Questions

**Comments to the Author**

1. Is the manuscript technically sound, and do the data support the conclusions?

Reviewer #1: Yes

Reviewer #2: Yes

2. Has the statistical analysis been performed appropriately and rigorously? 

Reviewer #1: Yes

Reviewer #2: I Don't Know

3. Have the authors made all data underlying the findings in their manuscript fully available?

Reviewer #1: Yes

Reviewer #2: Yes

4. Is the manuscript presented in an intelligible fashion and written in standard English?

Reviewer #1: Yes

Reviewer #2: Yes

5. Review Comments to the Author

Reviewer #1: U-Shaped association between serum triglyceride levels and mortality among septic patients: an analysis based on the MINIC-IV database is a research paper that studied the role of triglyceride levels in mortality of sepsis patients.

My suggestions are as follows:

- The link between triglyceride and mortality of sepsis patients must be discussed in inflammation basis. It is well known that infections (Acta Medica Mediterranea, 2013, 29: 551-554) and sepsis (Frontiers in cell and developmental biology, 2019, 7: 108) are associated with inflammation. Moreover, triglyceride mbased markers are also associated with inflmmatory conditions such as type 2 DM (Acta facultatis medicae Naissensis, 2022, 39.1: 66-73. DOI: 10.5937/afmnai39-33239), and liver steatosis (Experimental Biomedical Research, 2021, 4.3: 224-229. DOI:10.30714/j-ebr.2021370081). Thus, triglyceride levels may be associated with sepsis mortality via inflammation pathway. Discuss please.

- Can the authors replace figures with new versions which have higher resolution?

- Can authors state significant p values as bold in the tables?

Reviewer #2: Title:

U-Shaped association between serum triglyceride levels and mortality among septic

patients: An analysis based on the MINIC-IV database

Summary

This paper presents results that seek to investigate the association between serum TG levels and mortality in septic patients admitted to the intensive care unit. The study is commendable and provides more data to support the exact relationship between TG levels and mortality in septic patients where results from prior studies have been largely conflicting. The rationale for the present study is clear and the problem was well stated. The reviewer recommends that some minor yet important revisions of the manuscript be carried out.

Minor revisions

• Limitations of the study have been stated exhaustively. What about the strengths of the study?

• Page 5 lines 72-79 are methodology and results. This should NOT be part of the background/introduction of the study.

• Page 5 line 85. MIT should be written in full at the first mention.

• Page 6 line 117. SOFA should be written in full at the first mention

• Page 8 lines 141-142 are NOT methods but rather results. Remove or send to the results section

• Page 9 lines 175-184 are NOT results. They should be part of the methodology section.

6. PLOS authors have the option to publish the peer review history of their article (what does this mean?). If published, this will include your full peer review and any attached files.

Reviewer #1: No

Reviewer #2: No

---

## [Author Response · Author response to Decision Letter 0]

17 Aug 2023

Reviewer #1:

U-Shaped association between serum triglyceride levels and mortality among septic patients: an analysis based on the MIMIC-IV database is a research paper that studied the role of triglyceride levels in mortality of sepsis patients.

My suggestions are as follows:

Response: On behalf of my co-authors, we thank you very much for your consideration of our manuscript entitled “U-Shaped association between serum triglyceride levels and mortality among septic patients: an analysis based on the MIMIC-IV database” (Manuscript ID: PONE-D-23-19155). We greatly appreciate your comments. We have revised the manuscript accordingly, indicating the changes in the manuscript text file with blue. These changes have significantly strengthened and improved the manuscript.

1. The link between triglyceride and mortality of sepsis patients must be discussed in inflammation basis. It is well known that infections (Acta Medica Mediterranea, 2013, 29: 551-554) and sepsis (Frontiers in cell and developmental biology, 2019, 7: 108) are associated with inflammation. Moreover, triglyceride based markers are also associated with inflmmatory conditions such as type 2 DM (Acta facultatis medicae Naissensis, 2022, 3 9.1: 66-73. DOI: 10.5937/afmnai39-33239), and liver steatosis (Experimental Biomedical Research, 2021, 4.3: 224-229. DOI:10.30714/j-ebr.2021370081). Thus, triglyceride levels may be associated with sepsis mortality via inflammation pathway. Discuss please.

Response: We appreciate the reviewer’s comment.

It is indeed that triglyceride levels may be associated with sepsis mortality via inflammation pathway. Following your suggestion, we have performed a discussion in the revised manuscript as following:

“Recent studies have pointed out that TG-based markers are associated with many inflammatory conditions including type 2 diabetes mellitus, coronary heart disease and liver steatosis [41-43]. The initial phase of sepsis is characterized by overwhelming inflammation [44]. Interestingly, a recent study revealed that a combination of bis (α-furancarboxylato) oxovanadium and metformin ameliorates hepatic steatosis in high-fat diet-induced obese mice by alleviating hepatic inflammation and enhancing the insulin signaling pathway [45]. Thus, we speculate that TG may be associated with sepsis mortality via the inflammation pathway. This should be further investigated soon.”

2. Can the authors replace figures with new versions which have higher resolution?

Response: Thanks for raising this important point. 

All the figures had been redrawn and replaced in the revised manuscript. The resolution of the new figures is 600dpi.

3. Can authors state significant p values as bold in the tables?

Response: Thanks for the reviewer’s kind reminder. 

Following the reviewer’s suggestion, the significant p values in the revised manuscript had been adjusted as bold in the tables.

Reviewer #2: 

Title: U-Shaped association between serum triglyceride levels and mortality among septic patients: An analysis based on the MINIC-IV database

Summary: This paper presents results that seek to investigate the association between serum TG levels and mortality in septic patients admitted to the intensive care unit. The study is commendable and provides more data to support the exact relationship between TG levels and mortality in septic patients where results from prior studies have been largely conflicting. The rationale for the present study is clear and the problem was well stated. The reviewer recommends that some minor yet important revisions of the manuscript be carried out.

Response: Thanks so much for your comments concerning our manuscript entitled “U-Shaped association between serum triglyceride levels and mortality among septic patients: an analysis based on the MIMIC-IV database” (Manuscript ID: PONE-D-23-19155). Those comments are all valuable and very helpful for revising and improving our paper. We have studied your comments carefully and made substantial revisions. We hope that these corrections will meet with approval.

Minor revisions

1. Limitations of the study have been stated exhaustively. What about the strengths of the study?

Response: Thanks for pointing out this issue. 

This is the first retrospective observational study to suggest a U-shaped association between serum TG levels in septic ICU patients is 300-500 mg/dL. According to our results, we should maintain serum TG levels in septic patients at 300-500 mg/dL.

2. Page 5 lines 72-79 are methodology and results. This should NOT be part of the background/introduction of the study.

Response: Thanks for pointing out this issue. 

The main purpose of this paragraph is to give a brief summary of the results of our study, so as to facilitate the reader's reading. Following your suggestion, the paragraph had been re-edited in revised manuscript. Details were as following: 

“Here, a total of 2,782 septic patients included in an online public database called “Multiparameter Intelligent Monitoring in Intensive Care IV” (MIMIC-IV) were studied. We found a U-shaped association between serum TG level and the risk of mortality. The lowest risk of mortality was associated with a serum TG level of 300-500 mg/dL.”

3. Page 5 line 85. MIT should be written in full at the first mention.

Page 6 line 117. SOFA should be written in full at the first mention

Response: Thanks for your kind reminder. 

Following the reviewer’s suggestion, this information has been provided in the revised manuscript.

5. Page 8 lines 141-142 are NOT methods but rather results. Remove or send to the results section

Response: Thanks for your suggestion.

The sentence was removed in the revised manuscript.

6. Page 9 lines 175-184 are NOT results. They should be part of the methodology section.

Response: Thanks for your suggestion.

This paragraph is mentioned in the "Patient inclusion" section of the “Methods”. Thus, the paragraph was removed in the revised manuscript.

---

## [Decision Letter · Decision Letter 1]

9 Nov 2023

U-shaped association between serumtriglyceride levels and mortality among septic patients: an analysis based on the MIMIC-IV database

PONE-D-23-19155R1

Dear Dr.Dadong Liu,

We’re pleased to inform you that your manuscript has been judged scientifically suitable for publication and will be formally accepted for publication once it meets all outstanding technical requirements.

Kind regards,

Paavani Atluri

Academic Editor

PLOS ONE

Additional Editor Comments (optional):

Reviewers' comments:

Reviewer's Responses to Questions

**Comments to the Author**

1. If the authors have adequately addressed your comments raised in a previous round of review and you feel that this manuscript is now acceptable for publication, you may indicate that here to bypass the “Comments to the Author” section, enter your conflict of interest statement in the “Confidential to Editor” section, and submit your "Accept" recommendation.

Reviewer #1: All comments have been addressed

Reviewer #2: All comments have been addressed

2. Is the manuscript technically sound, and do the data support the conclusions?

Reviewer #1: Yes

Reviewer #2: Yes

3. Has the statistical analysis been performed appropriately and rigorously? 

Reviewer #1: Yes

Reviewer #2: Yes

4. Have the authors made all data underlying the findings in their manuscript fully available?

Reviewer #1: Yes

Reviewer #2: Yes

5. Is the manuscript presented in an intelligible fashion and written in standard English?

Reviewer #1: Yes

Reviewer #2: Yes

6. Review Comments to the Author

Reviewer #1: My criticisms are appropriately addressed. revised auricle is free from flaws. i think it can be accepted.

Reviewer #2: The authors have responded to all queries and incorporated all recommendations. The manuscript has been improved, and is now suitable for publication in PLOS ONE

7. PLOS authors have the option to publish the peer review history of their article (what does this mean?). If published, this will include your full peer review and any attached files.

Reviewer #1: No

Reviewer #2: No

---

## [Editor Report · Acceptance letter]

14 Nov 2023

PONE-D-23-19155R1 

U-shaped association between serum triglyceride levels and mortality among septic patients: an analysis based on the MIMIC-IV database 

Dear Dr. Liu:

I'm pleased to inform you that your manuscript has been deemed suitable for publication in PLOS ONE. Congratulations! Your manuscript is now with our production department. 

Kind regards, 

on behalf of

Dr. Paavani Atluri 

Academic Editor

PLOS ONE